# Ion Channels as Potential Drug Targets in Dry Eye Disease and Their Clinical Relevance: A Review

**DOI:** 10.3390/cells13232017

**Published:** 2024-12-06

**Authors:** Carl Randall Harrell, Vladislav Volarevic

**Affiliations:** 1Regenerative Processing Plant LLC, 13700 Reptron Blvd, Tampa, FL 33626, USA; 2Center for Research on Harmful Effects of Biological and Chemical Hazards, Departments of Genetics, Microbiology and Immunology, Faculty of Medical Sciences, University of Kragujevac, 69 Svetozar Markovic Street, 34000 Kragujevac, Serbia; drvolarevic@yahoo.mail; 3Faculty of Pharmacy Novi Sad, Heroja Pinkija 4, 21000 Novi Sad, Serbia

**Keywords:** sodium chloride channels, natural cytokine communication, hypo-osmotic solutions, dry eye disease, tear film homeostasis, d-MAPPS™ Hypo-Osmotic Ophthalmic Solution

## Abstract

Dry eye disease (DED) is a common multifactorial disorder characterized by a deficiency in the quality and/or quantity of tear fluid. Tear hyperosmolarity, the dysfunction of ion channel proteins, and eye inflammation are primarily responsible for the development and progression of DED. Alterations in the structure and/or function of ion channel receptors (transient receptor potential ankyrin 1 (TRPA1), transient receptor potential melastatin 8 (TRPM8), transient receptor potential vanilloid 1 and 4 (TRPV1 and TRPV4)), and consequent hyperosmolarity of the tears represent the initial step in the development and progression of DED. Hyperosmolarity triggers the activation of ion channel-dependent signaling pathways in corneal epithelial cells and eye-infiltrated immune cells, leading to the activation of transcriptional factors that enhance the expression of genes regulating inflammatory cytokine production, resulting in a potent inflammatory response in the eyes of DED patients. A persistent and untreated detrimental immune response further modifies the structure and function of ion channel proteins, perpetuating tear hyperosmolarity and exacerbating DED symptoms. Accordingly, suppressing immune cell-driven eye inflammation and alleviating tear hyperosmolarity through the modulation of ion channels in DED patients holds promise for developing new therapeutic strategies. Here, we summarize current knowledge about the molecular mechanisms responsible for the inflammation-induced modification of ion channels leading to tear hyperosmolarity and immune cell dysfunction in DED patients. We also emphasize the therapeutic potential of the newly designed immunomodulatory and hypo-osmotic solution d-MAPPS™ Hypo-Osmotic Ophthalmic Solution, which can activate TRPV4 in corneal epithelial cells, stabilize the tear film, enhance natural cytokine communication, and suppress detrimental immune responses, an important novel approach for DED treatment.

## 1. Introduction

Dry eye disease (DED) is a common eye condition affecting over half the world’s population [1]. This multifactorial disorder, affecting both the lacrimal system and the ocular surface, is characterized by a deficiency in the quality and/or quantity of tear fluid [1]. DED has two primary subtypes: evaporative dry eye (EDE), caused by excessive evaporation of tear fluid, and aqueous tear-deficient dry eye (ADDE), characterized by the lacrimal glands’ inefficiency or failure to produce tears [2,3]. In clinical settings, DED is often seen as a “combination” or “hybrid” form of these two types, where each type gains certain clinical characteristics from the other, initiating and worsening its pathology [2,3].

Many factors, including aging, eye injury, surgical interventions, poor nutrition, environmental allergens, and chemical hazards, can cause dysfunction of ion channel proteins and generate a detrimental immune response, leading to the development and progression of DED [4,5]. Tear hyperosmolarity (caused by alterations in ion channels) and ocular inflammation (caused by enhanced activation of immune cells and massive production of inflammatory cytokines) create a vicious cycle, aggravating all main DED-related signs and symptoms (ocular discomfort, pain, and vision problems) [1,5,6].

Accordingly, many experimental and clinical studies have focused on examining the ion channel-dependent effects on the development of DED-associated pathology [1,3,4,5,6]. The delineation of signaling pathways which were elicited in eye-infiltrated immune cells during DED progression has paved the way for designing new immunoregulatory therapeutic agents capable of suppressing detrimental immunity, attenuating DED-related signs and symptoms [1].

This manuscript summarizes current knowledge about the molecular mechanisms responsible for inflammation-induced modification of ion channels leading to tear hyperosmolarity and immune cell dysfunction in DED patients. The therapeutic potential of d-MAPPS™, the newly designed immunomodulatory and hypo-osmotic ophthalmic solution, which can stabilize the tear film, enhance natural cytokine communication, and suppress detrimental immune responses, is also emphasized. An extensive literature review was carried out in June 2024 across several databases (MEDLINE, EMBASE, and Google Scholar), from 1990 to present. Keywords used in the selection were as follows: ”dry eye disease”, “tear hyperosmolarity”, “ion-channels”, “meiboman gland dysfunction”, “signaling pathways”, “inflammation”, “immune cells”, “immunoregulation”, “hypo-osmotic ophthalmic solution”. All journals were considered and an initial search retrieved 104 articles. The abstracts of all these articles were reviewed to check for their relevance to the subject of this manuscript. Eligible studies had to delineate molecular and cellular mechanisms that are responsible for the dysfunction of ion channels in the eyes of DED patients and their findings were analyzed in this review.

## 2. The Impact of Ion Channels on DED Development and Progression

Ion channels can be classified based on their selectivity for a specific type of ion (selective or non-selective channels) and dependence on membrane potential (voltage-dependent and voltage-independent channels) [7]. Voltage-dependent ion channels are characterized by their response to changes in membrane potential, playing critical roles in excitability, while voltage-independent channels operate based on alternative stimuli (changes in osmolarity, heat, cold, pH, etc.) [7]. Both voltage-dependent and voltage-independent ion channels play an important role in maintaining ocular surface homeostasis [8,9,10,11]. Voltage-gated sodium channels are found in corneal neurons and their dysfunction or altered expression can provoke heightened sensitivity, pain, and discomfort in the eyes of DED patients [8,9,10,11]. Voltage-gated potassium channels maintain the resting membrane potential and their dysfunction may alter tear secretion, contributing to the aggravation of DED-related symptoms [9,10,11]. Voltage-gated calcium channels are found in lacrimal gland cells where regulate calcium influx, promoting tear production [8,12,13]. Accordingly, dysfunction of these channels results in impaired calcium signaling which can lead to decreased tear secretion and DED development [12,13].

Partially voltage-dependent ion channels such as transient receptor potential melastatin 8 (TRPM8) and voltage-independent ion channels (transient receptor potential ankyrin 1 (TRPA1) and transient receptor potential vanilloid 1 and 4 (TRPV1 and TRPV4)) belong to the transient receptor potential (TRP) family which comprises a diverse group of ion channels that play critical roles in sensory perception, particularly in nociception, temperature sensation, and osmoregulation [14]. These ion channels maintain cellular ion homeostasis and are intimately involved in the regulation of ocular surface health [14]. Specific TRP channels exhibit distinct expression patterns across various cell layers and types within the eyes [15]. For instance, TRPV1 and TRPA1, both of which are involved in pain and irritation sensation, are predominantly expressed in excitable cells such as corneal sensory neurons [15,16]. These neurons detect environmental stimuli and contribute to the corneal pain response, especially under conditions of dryness or inflammation [15]. In contrast, non-excitable cells like epithelial and goblet cells express TRPV4, which is involved in mechanosensation and contributes to the regulation of tear secretion and homeostasis [17]. TRPV4 is also expressed on immune cells, playing an important role in the regulation of immune cell-driven corneal injury and inflammation [18]. TRPM8, known for its role in cold sensation, is found in the corneal epithelium, enhancing the eye’s response to temperature changes. The differential expression of these TRP channels across excitable and non-excitable cells underscores their multifaceted roles in maintaining ocular surface health.

TRPA1 controls the influx of calcium cations through the cell membrane [1,11]. It is highly expressed in the trigeminal nerve [1,11]. The activation of TRPA1 in an animal model of DED resulted in decreased tear production, more frequent blinking, and forelimb eye wipe behavior [11,19].

The TRPM8 channel, known for its role as a cold- and menthol-sensitive ion channel, exhibits distinct functions in neuronal and non-neuronal cells within the eyes [12,20,21]. In neuronal cells, such as those located in the cornea and conjunctiva, TRPM8 primarily acts as a sensory receptor that detects changes in temperature and osmotic pressure [22]. When activated by cold temperatures or menthol, TRPM8 channels facilitate the influx of cations like calcium and sodium, leading to the depolarization of sensory neurons [22,23]. This process is crucial for generating sensations of coolness and the reflex tearing response, which helps protect and lubricate the ocular surface [12,22]. Accordingly, when the cornea experiences dryness or irritation, TRPM8 activation can trigger the production of tears, thereby enhancing comfort and maintaining eye health [12,22]. In non-neuronal cells, TRPM8 plays a role in regulating tear production and osmotic balance [24]. TRP8-expressing non-neuronal cells respond to osmotic changes by utilizing TRPM8 to maintain hydration and support mucin secretion, which is vital for forming a stable tear film [24]. When the tear film becomes hyperosmolar, TRPM8 can activate intracellular signaling pathways that promote the secretion of protective mucins, thus contributing to the overall health of the ocular surface [24]. Therefore, TRPM8 serves as a critical mediator in sensory neuronal cells for detecting environmental changes and initiating protective reflexes, while in non-neuronal cells, it helps regulate tear production and maintain osmotic homeostasis, underscoring its multifaceted role in ocular physiology [12,22,23,24]. In the eyes of DED patients, the chronic exposure to hyperosmolar conditions leads to the dysregulation of TRPM8 channels, resulting in reduced sensitivity and impaired reflex tearing [12,25]. This impairment contributes to a cycle of dryness and discomfort, exacerbating inflammation and further compromising tear film stability [12,25]. The question of whether a TRPM8 agonist or antagonist is more effective for treating DED remains unresolved [25]. On the one hand, TRPM8 agonists like menthol produced a cooling effect and heightened tear production in a mouse model of DED, but varying dosages also resulted in heightened pain-related behaviors [25]. On the other hand, selective TRPM8 antagonists, including AMTB and BCTC, have been shown to reduce the activity of the orbicularis oculi muscle and alleviate corneal dryness triggered by hypertonic saline [25].

TRPV1 is found on sensory neurons, playing a crucial role in pain perception and neurogenic inflammation [6,15,16,26,27]. The activation of TRPV1 in corneal epithelial cells leads to the release of pro-inflammatory factors such as IL-6 and IL-8 [28]. Chronic DED animal models show increased TRPV1 expression in both corneal nerve fibers and the trigeminal ganglia, suggesting a link between TRPV1 and DED development [29,30]. Additionally, it was recently demonstrated that both TRPM8 and TRPV1 are expressed at the gene and protein levels in corneal epithelia and meibomian glands where they influence meibocyte lipogenesis and, therefore, may serve as potential drug targets in the treatment of meibomian gland dysfunction (MGD) and DED [8,31,32]. Also, it should be noted that both TRPM8 and TRPV1 channels interact with sodium and potassium channels, influencing their expression [6,14]. Blocking TRPM8 in DED animal models reduced the mRNA levels of sodium channels Nav1.7 and NaV1.8 [6,32]. Similarly, blocking TRPV1 reduced the mRNA levels of these sodium channels [6,32].

TRPV4 is crucial for mechanosensation, osmosensation, ion homeostasis, inflammatory responses, barrier integrity, and wound healing in human corneal epithelial cells [17,18]. It is sensitive to the alterations in osmolarity and responds to the changes in osmotic pressure and stress, playing an important role in maintaining corneal transparency and integrity [33,34]. In corneal epithelial cells, TRPV4 is involved in the maintenance of tight junctions, enhancing the barrier function of the epithelium [34]. Following corneal injury, TRPV4 activation can influence healing processes by modulating viability and the proliferation of injured corneal epithelial cells [34,35]. The activation of TRPV4 leads to an influx of calcium ions which, through the interaction with several kinases and transcriptional factors, increase the expression of genes that prevent apoptosis and stimulate the proliferation and migration of corneal epithelial cells, enhancing repair and regeneration of injured epithelium [34,35].

In addition to TRP channels, purinergic receptors (P2X and P2Y), aquaporins (AQPs), cystic fibrosis transmembrane conductance regulator (CFTR), inositol trisphosphate (IP3), and ryanodine receptors also regulate tear osmolarity, maintaining ocular surface health [36,37]. Purinergic receptors are activated by extracellular ATP, a molecule released during tissue injury and inflammation. P2X receptors are found in increased numbers on immune cells at sites of inflammation, and their activation triggers the release of inflammatory cytokines, exacerbating the inflammatory response [36]. AQPs are transmembrane channels which facilitate water and small-solute transport across cell membranes [36]. AQP5 is essential for lacrimal gland fluid secretion [36]. Hyperosmotic stress triggers increased AQP5 mRNA expression in human corneal epithelial cells, promoting transepithelial fluid efflux and protecting corneal barrier function during tear hyperosmolarity [36]. CFTR is a cAMP-activated chloride anion channel expressed in corneal and conjunctival epithelial cells [37]. CFTR activation significantly improved tear secretion in both the healthy and inflamed eyes of DED animal models, highlighting its potential as a therapeutic target for DED [37].

In corneal and conjunctival cells, IP3 and ryanodine receptors serve as crucial intracellular calcium release channels, playing a pivotal role in cellular signaling and physiological responses [38,39,40]. When membrane receptors are activated, the enzyme phospholipase C generates IP3 from phosphatidylinositol 4,5-bisphosphate, leading to the release of calcium ions from the endoplasmic reticulum (ER) through IP3 receptors [40]. This rise in intracellular calcium is essential for various cellular processes, including tear secretion, epithelial cell proliferation, and inflammatory responses [40]. Additionally, ryanodine receptors, which are also present in these cells, further contribute to the dynamics of calcium ions by releasing calcium from the ER in response to localized calcium influx or other signaling events, amplifying the calcium signal [39]. This coordinated action of IP3 and ryanodine receptors helps regulate the excitability of sensory neurons in the cornea and modulates the function of conjunctival goblet cells, influencing mucin secretion essential for maintaining a stable tear film and ocular surface health [38,39,40]. These findings suggest that alterations in the structure and/or function of TRP channels, IP3, and ryanodine receptors and consequent hyperosmolarity of the tears represent the initial step in the development and progression of DED [39,40].

## 3. Signaling Pathways Responsible for Hyperosmolarity-Dependent Inflammatory Response in DED Patients

TRPA1 and TRPV1 ion channels play significant roles in modulating tissue inflammatory and immune responses, often relying on extensive interactions between sensory neurons and immune cells [26]. A higher propensity of TRPV1-expressing corneal sensory nerves was observed after corneal injury, along with their interaction with intra-epithelial leukocytes and professional antigen-presenting dendritic cells (DCs) [41].

A low-grade inflammatory response to tissue stress, occurring during lens wear in humans, leads to alterations in the number of intra-epithelial DCs [42]. Both TRPA1 and TRPV1 ion channels are essential for contact lens-induced corneal parainflammatory responses and maintaining baseline levels of certain resident corneal immune cells [42].

Hyperosmolarity triggers the activation of the TRPV1-dependent signaling pathway in corneal epithelial cells, leading to the activation of the transcriptional factor NF-κB [6,43]. This causes an increase in the expression of genes for TNF-α and IL-1β, resulting in enhanced production of these inflammatory cytokines. IL-1β and TNF-α stimulate various kinases (mixed lineage kinase (MLK)-2/-3, transforming growth factor β-activated kinase 1 (TAK1), mitogen-activated protein kinase (MKK)-1, apoptosis signal-regulating kinase 1 (ASK1), MKK7, and Jun N-terminal kinase (JNK)) in eye-infiltrated, professional antigen-presenting DCs [44,45]. These kinases phosphorylate and activate transcription factors c-Jun and ATF2. Activated c-Jun and ATF2 move to the nucleus, where they increase the expression of pro-Th1 (IL-12) and pro-Th17 cytokines (IL-1β, IL-6, IL-23) in DCs [44,45]. Through the production of IL-12, DCs stimulate signal transducers and activators of (STAT)-1 and T-bet transcriptional factors in naïve CD4+ T lymphocytes, leading to their differentiation into effector, interferon gamma (IFN-γ)-producing Th1 cells. In a similar manner, by stimulating STAT-3 and RAR-related orphan receptor gamma T (RORγT) transcriptional factors in naïve CD4+T cells, DC-derived IL-1β, IL-6, and IL-23 promote the development and expansion of inflammatory CD4+Th17 cells [44,45].

In the eyes of DED patients, Th1 cell-derived interferon gamma (IFN-γ), Th17 cell-sourced IL-17, and IL-22 induce the generation of inflammatory M1 phenotype in macrophages and N1 phenotype in neutrophils, crucially contributing to the generation of innate cell-driven inflammation [46,47,48]. N1 neutrophils and M1 macrophages are valuable sources of inflammatory mediators, which contribute to the reduced expression of mucins, apoptotic death of corneal and conjunctival epithelial cells, and to the loss of goblet cells in the inflamed eyes of DED patients [46]. Additionally, M1 macrophages and N1 neutrophils produce Th1 and Th17 cell-attracting chemokines (C-X-C motif ligand (CXCL)-9/-10 and C-C motif ligand (CCL)-20), enabling massive influxes of CD4+ and CD8+ Th1 and Th17 cells in the inflamed eyes of DED patients [46,47,48].

Th1 and Th17 lymphocytes, produced by generating inflammatory cytokines (IFN-γ and IL-17), cause damage to the corneal epithelial barrier, trigger inflammation of the meibomian glands, and reduce the production of tears in DED mice [47,48]. Additionally, blocking antibody-dependent neutralization or genetic removal of IFN-γ and IL-17 significantly reduces eye inflammation and almost completely stops damage to the corneal epithelial barrier in DED mice, proving the essential role of these inflammatory cytokines in DED development [47,48]. IL-17 produced by Th17 cells increases the expression of MMP-3 and MMP-9, which in turn leads to the breakdown of the corneal epithelial barrier [46,47].

Additionally, Th1 cell-derived IFN-γ activates both extrinsic and intrinsic apoptotic pathways in corneal epithelial cells [48]. The extrinsic or “death receptor” pathway involves the interaction between the Fas ligand (FasL)- and TNF-related apoptosis-inducing ligand (TRAIL)-activated caspase-8, while the intrinsic or “mitochondrial” pathway is activated by caspase-9 [46]. While the administration of exogenous IFN-γ significantly increased caspase-3-, -8-, and -9-dependent apoptosis in conjunctival epithelial cells, the pharmacological inhibition or genetic deletion of IFN-γ significantly reduced immune cell-induced damage to the conjunctival epithelial and goblet cells in DED mice [46,48]. Similar outcomes have been observed in DED patients; a disturbance in the corneal barrier associated with IL-17 was found to increase the likelihood of corneal ulcers and vision loss, whereas higher levels of IFN-γ were linked to the development of conjunctival epithelial squamous metaplasia, resulting in a progressive loss of goblet cells [46,47,49].

## 4. Tear Hyperosmolarity and the Vicious Cycle in DED

Severe DED develops due to enhanced Th1 and Th17 cell-driven eye inflammation, resulting in pain due to the primary stimulation and sensitization of polymodal nociceptors in the inflamed eyes [47,48]. Chronic inflammation leads to long-lasting alterations in the expression and function of stimulus-transducing and voltage-sensitive sodium and calcium ion channels, modifying the excitability of polymodal terminal receptors and evoking chronic inflammatory pain [46].

Injured cells within the inflamed eye and activated immune cells secrete various bioactive factors, including hydrogen ions (H+), adenosine triphosphate (ATP), adenosine, protons, prostaglandins, leukotrienes, bradykinin, serotonin, platelet-activating factor, histamine, and inflammatory cytokines, which stimulate spontaneous impulse activity in nociceptor terminals, reducing the threshold for stimulus detection and increasing the amplitude of nerve impulse responses elicited by suprathreshold stimulation [50].

The sensitization of peripheral nociceptors is responsible for the unique quality and persistence of pain experienced from inflamed tissues [50]. Inflammatory factors, including prostaglandins (PGE2), bradykinin, serotonin, TNF-α, IL-1β, IL-6, and IL-8, bind to their respective receptors (neurotrophic tyrosine kinase receptors (NTKR), including TrkA, and/or G-protein coupled receptors (GPCRs), such as bradykinin and PGE2 receptors, and cytokine receptors), activating multiple intracellular pathways [46]. These pathways include protein kinase A (PKA), protein kinase B (PKB), protein kinase C (PKC), mitogen-activated protein kinase (MAPK), and the Ca++/Calmodulin-dependent kinase I and II (CamKI/II). The activation of these signaling pathways results in the post-translational modification of ion channel proteins, affecting their function [51].

Post-translational modifications exert a broad range of effects on ion channel proteins, varying from highly stable modifications to those that are transient and reversible [51]. For example, glycosylation and the formation of disulfide bonds are directly involved in the synthesis, maturation, and folding of proteins [51]. The covalent binding of molecules, such as the addition of a ubiquitin moiety, facilitates rapid and stable modifications. Furthermore, the incorporation of a phosphoryl group, which carries two negative charges at physiological pH, can alter the protein’s structure and function by modifying the free energy landscape [51].

Post-translational modifications of Navs are crucial for the development of tear-hyperosmolarity-dependent DED [51]. Navs are activated in response to the depolarization of the transmembrane voltage, leading to the generation of a fast, transient, and significant inward sodium current, which is responsible for the ascending phase of the action potential [51]. Navs contain charged residues in the voltage sensor domain that are sensitive to changes in membrane potential. Upon a change in transmembrane voltage, these charged domains shift in the electric field, resulting in conformational changes known as gating [51]. The addition of charged groups to their intracellular, extracellular, or transmembrane domains modifies the intrinsic properties and functions of the protein. In addition to the direct electrostatic influence on gating, phosphorylation can also create or disrupt binding sites for interaction with other regulatory proteins that affect Nav function [51].

The molecular mechanisms underlying inflammation-driven post-translational modifications of Navs involve the activation of various intracellular kinases [52,53]. The binding of PGE2 and bradykinin to GPCRs predominantly leads to the activation of PKA and PKC kinases through the adenylate cyclase (AC) and inositol 3-phosphate (IP3) secondary messengers, respectively [52,53]. The binding of NTKR and CR to TNF-α, IL-6, and NGF activates ERK1/2 and p38 kinases via different potential secondary messengers. These signal transduction pathways can also undergo crosstalk with one another [52,53]. The increased activation of kinases, the downregulation of Nedd4-2, and the accumulation of methylglyoxal result in enhanced function or expression of Nav1.7/Nav1.8 channels at the cell membrane, leading to increased sodium influx and consequently to nociceptive neuronal hyperexcitability [52,53]. In a similar manner, potassium channels may also be activated because of inflammation-induced post-translational modifications of protein subunits [9].

Therefore, by inducing structural and functional changes in ion channels, a long-lasting and untreated detrimental immune cell-driven inflammatory response causes hyperosmolarity of the tears and generates a vicious cycle in the eyes, aggravating all DED-associated signs and symptoms [1].

## 5. The Therapeutic Potential of d-MAPPS™ Hypo-Osmotic Ophthalmic Solution in the Attenuation of Inflammation and Hyperosmolarity-Induced Pathological Changes in the Eyes of DED Patients

d-MAPPS™ Hypo-Osmotic Ophthalmic Solution is a hypotonic ophthalmic solution, containing many immunoregulatory, angio-modulatory, and trophic factors, designed to efficiently attenuate ongoing eye inflammation and tear hyperosmolarity, promoting enhanced tissue repair and regeneration [54]. Specifically, d-MAPPS™ Hypo-Osmotic Ophthalmic Solution contains interleukin 1 receptor antagonist (IL-1Ra), soluble receptors of tumor necrosis factor alpha (sTNFRI, sTNFRII), growth-related oncogene gamma (GRO-γ), fatty acid-binding protein 1 (FABP1), and platelet factor 4 (PF4), which alleviate eye inflammation, support tear stability, and prevent ocular surface epithelial damage, contributing to the enhanced repair and regeneration of the ocular surface epithelial barrier in DED patients [54].

As a hypotonic ophthalmic solution, d-MAPPS™ Hypo-Osmotic Ophthalmic Solution has been shown to effectively stabilize the tear film, thereby mitigating the symptoms of DED. It works by restoring the natural osmolarity of the tear film, reducing cellular stress, and indirectly supporting the body’s ability to regulate inflammation without the introduction of exogenous proteins or cytokines.

d-MAPPS™ Hypo-Osmotic Ophthalmic Solution functions by restoring the natural osmolarity of the tear film, a key factor in alleviating the cellular stress associated with DED. D-MAPPS™-dependent decrease in tear osmolarity may activate TRPV4 ion channel, allowing a massive influx of calcium ions into corneal epithelial cells [54]. The rise in intracellular calcium ions activates various signaling pathways that lead to the activation of potassium channels and other ion transporters [17]. In this way, d-MAPPS™-dependent activation of TRPV4 ion channels could importantly contribute to the process of regulatory volume decrease which would facilitate the restoration of osmotic balance. Importantly, TRPV4-driven activation of calcium-dependent signaling enhances expression of genes that prevent apoptosis and promote the proliferation of corneal epithelial cells [17]. In this way, d-MAPPS™-dependent activation of TRPV4 and consequent osmotic regulation could support healing processes in the eyes of DED patients by modulating viability, proliferation, and migration of injured corneal epithelial cells [54].

By stabilizing the tear film, this solution not only reduces the direct symptoms of irritation and dryness but also impacts the broader inflammatory pathways, reducing the need for exogenous anti-inflammatory agents. Among various bioactive factors, d-MAPPS ™ Hypo-Osmotic Ophthalmic Solution containing FABP1 protein is mainly responsible for these beneficial effects in the attenuation of tear hyperosmolarity [54,55]. FABP proteins regulate trans-epithelial water transport and maintain the epithelial barrier at the ocular surface [55]. Reduced expression and production of FABP proteins are associated with increased tear evaporation [55]. Since d-MAPPS™ Hypo-Osmotic Ophthalmic Solution contains a high concentration of FABP1 protein, it may regulate trans-epithelial water transport, support tear stability, and prevent ocular surface epithelial damage in the eyes of DED patients, resulting in the alleviation of dryness, grittiness, scratchiness, and soreness [54].

In addition to FABP, d-MAPPS™ Hypo-Osmotic Ophthalmic Solution contains IL-1Ra, sTNFRI, and sTNFRII, which directly suppress IL-1β- and TNF-α-driven recruitment of circulating leukocytes and TNF-α- and IL-1β-dependent alterations of ion channels, reducing hyperosmolarity and the total number of immune cells at the ocular surface [50,54,56]. When D-MAPPS™ Hypo-Osmotic Ophthalmic Solution containing IL-1Ra binds to the IL-1 receptor (IL-1R) on endothelial cells in the eyes of DED patients, the binding of IL-1β to IL-1R is blocked, and pro-inflammatory signals from IL-1R are stopped [54]. Consequently, various pro-inflammatory events initiated by IL-1β binding, including the synthesis and release of chemokines, enhanced influx of leukocytes, and post-translational modifications of ion channels in inflamed eyes, are inhibited by d-MAPPS™ Hypo-Osmotic Ophthalmic Solution containing IL-1Ra [54]. Similarly, d-MAPPS™ Hypo-Osmotic Ophthalmic Solution containing sTNFRI and sTNFRII binds to TNF-α and prevents TNF-α-dependent pathological events in the eyes of DED patients [54].

d-MAPPS™ Hypo-Osmotic Ophthalmic Solution effectively reduces the concentration of IL-12 in the supernatants of human peripheral blood mononuclear cells (pbMNCs) and reduces the production of IFN-γ in activated immune cells [57]. d-MAPPS™ Hypo-Osmotic Ophthalmic Solution containing GRO-γ inhibits DCs interactions and successfully attenuates the expansion of inflammatory Th1 immune cells that rely on DC-sourced IL-12. GRO-γ-treated DCs show a ‘tolerogenic’ feature with higher IL-10, lower IL-12, and IFN-γ production [57]. Therefore, it is highly expected that the intraocular administration of d-MAPPS™ Hypo-Osmotic Ophthalmic Solution affects the crosstalk between antigen-presenting IL-12-producing DCs and naive CD4+ T cells in lymph nodes, preventing DC-dependent generation of IFN-γ-producing Th1 lymphocytes. Since IFN-γ-producing T-cells increase the inflammatory capabilities of macrophages and push them towards inflammatory M1 cells, d-MAPPS™ Hypo-Osmotic Ophthalmic Solution inhibits the activation of intraocular M1 macrophages and reduces M1 macrophage-driven inflammation in the eyes of DED patients [54].

In line with these observations are findings obtained in pilot clinical trials, which showed that d-MAPPS™ Hypo-Osmotic Ophthalmic Solution successfully reduced eye discomfort and pain in 131 DED patients. Patients who received d-MAPPS™ Hypo-Osmotic Ophthalmic Solution experienced a substantial improvement in their symptoms, as evidenced by the significant drop in the visual analog scale (VAS) and standard patient evaluation of eye dryness (SPEED) scores three months after starting treatment, with the maximum improvement seen after a year of treatment. Additionally, d-MAPPS™ Hypo-Osmotic Ophthalmic Solution managed to promote the regeneration of injured meibomian glands and to efficiently attenuate DED-related symptoms in a patient suffering from MGD [58]. Importantly, d-MAPPS™ Hypo-Osmotic Ophthalmic Solution was well accepted by all patients, with no adverse reactions reported, indicating that the topical administration of d-MAPPS™ Hypo-Osmotic Ophthalmic Solution should be considered a new, safe, and efficacious method for treating MGD and DED [54,57,58].

## 6. Conclusions

Hyperosmolarity activates ion channel-dependent signaling pathways in corneal epithelial cells and eye-infiltrating immune cells, resulting in a massive release of alarmins and inflammatory cytokines which induce injury and inflammation in corneas, conjunctivas, lacrimal, and meibomian glands [59,60]. Prolonged and un-treated detrimental immune response can alter the structure and function of ion channel proteins, continuing the cycle of inflammation and tear-hyperosmolarity-dependent pathological changes in the eyes of DED patients [60]. Results obtained in experimental and pilot clinical trials demonstrated that the topical administration of d-MAPPS™, an immunoregulatory and hypo-osmotic ophthalmic solution, attenuated tear hyperosmolarity, suppressed detrimental inflammatory response, efficiently alleviated DED-related symptoms, and promoted the repair and regeneration of injured meibomian glands in DED patients [54,57,58]. Therefore, targeting immune cell-induced inflammation through natural cytokine modulation and attenuation of tear hyperosmolarity through the restoration of ion channels’ function should be explored in up-coming clinical studies as a novel and promising therapeutic approach which could offer hope for the development of more effective and sustainable DED management plans.

## Data Availability

Not applicable.

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
