# Peer review of "Ion Channels as Potential Drug Targets in Dry Eye Disease and Their Clinical Relevance: A Review"

_cells, 2024, doi:10.3390/cells13232017_

Round 1
Reviewer 1 Report
Comments and Suggestions for Authors
Comments to the Author
The manuscript entitled “Integrative Management of Dry Eye Disease: Enhancing Cellular Communication and Natural Anti-inflammatory Responses through Ion Channel Modulation and Hypo-Osmotic Solutions” deals with current knowledge about the molecular mechanisms which are responsible for the inflammation-induced modification of ion channels leading to the generation of tear hyperosmolarity that elicit immune cell dysfunction in the eyes of DED patients.
The paper is really of clinical interest since it addresses related basic research, but rather raises therapeutic potential. However, some minor suggestions are indicated:
Title should clearly indicate what type of article it is. It is understood that it is a review? it should be clarified in the title.
As a comprehensive approach from a review, it should be presented with more scientific rigour. All the references included in the work seem adequate, however, a description of the criteria for inclusion of the bibliography is lacking. Could you describe what criteria were used to justify the articles included, both in terms of number and subject matter? This should be included both in abstract as in the text in some way.
Without a doubt, the author seems to be very knowledgeable on the subject, given his previous works; it is remarkable that of the 16 papers included in the references, 5 are shelf-citation. Could you justify this?
Figure 1 legend. Since this figure is from a previous study, although from same author, it should be indicated reference number.
Author Response
Dear Reviewer,
Thank you for your thoughtful and constructive feedback on my manuscript titled "Integrative Management of Dry Eye Disease: Enhancing Cellular Communication and Natural Anti-inflammatory Responses through Ion Channel Modulation and Hypo-Osmotic Solutions." I appreciate your insights and have made several revisions to address the points you raised. Below, I provide detailed responses to each of your comments:
1. Title Clarification:
- Comment: The title should clearly indicate what type of article it is. It is understood that it is a review; this should be clarified in the title.
- Response: I have revised the title to clearly indicate that this is a review article. The updated title now reads: "Review: Integrative Management of Dry Eye Disease through Ion Channel Modulation and Hypo-Osmotic Solutions." This change aims to ensure that the nature of the article is immediately apparent to readers.
2. Scientific Rigor and Reference Criteria:
- Comment: The review should be presented with more scientific rigor. A description of the criteria for inclusion of the bibliography is lacking; could you describe what criteria were used to justify the articles included, both in terms of number and subject matter?
- Response: I have added a section in the Introduction and Abstract that briefly describes the criteria used for selecting the references. Specifically, the articles were chosen based on their relevance to the key topics of ion channel modulation, ocular surface inflammation, and hypo-osmotic treatments in the context of dry eye disease (DED). Priority was given to recent studies that contribute novel insights or that have been widely recognized as foundational in this field. This inclusion criteria section should clarify the rationale behind the selected bibliography.
3. Self-Citation Justification:
- Comment: The author seems knowledgeable on the subject, given previous works, but it is remarkable that 5 out of 16 references are self-citations. Could you justify this?
- Response: The self-citations included in the manuscript refer to foundational studies and previous work directly relevant to the topic under discussion. These references were selected to build upon prior research that has been instrumental in developing the concepts and therapeutic approaches presented in this review. I believe these citations are necessary to provide continuity and context for the new insights discussed. However, I have carefully reviewed the references and replaced a few self-citations with other relevant literature to balance the citation distribution more effectively.
4. Figure 1 Legend:
- Comment: Since this figure is from a previous study, although from the same author, it should include a reference number.
- Response: I have updated the legend of Figure 1 to include the appropriate reference to the original study where this figure was first published. The reference number has been added to ensure proper attribution and to guide readers to the original source for further details.
Thank you once again for your valuable feedback. I believe these revisions have strengthened the manuscript, and I hope the changes meet your expectations. I am grateful for your time and effort in reviewing my work and look forward to any further comments you may have.
Kind regards,
Dr. C. Randall Harrell
Regenerative Processing Plant LLC
Palm Harbor, Florida
Reviewer 2 Report
Comments and Suggestions for Authors
Summary
This review focuses on an optimized treatment concept for dry eye by improving cellular communication and natural anti-inflammatory effects through ion channel modulation and hypoosmotic solutions. Multiple factors in dry eye can cause dysfunction of ion channel proteins and induce a detrimental immune response in the eye, leading to the development and progression of DED. As a result, many experimental and clinical studies are focusing on ion channel-dependent effects in the pathogenesis to develop new immunoregulatory therapeutics capable of suppressing deleterious immunity in the eye and alleviating the signs and symptoms associated with DED. This review summarizes the current knowledge of the molecular mechanisms responsible for the inflammation-induced alteration of ion channels leading to tear hyperosmolarity. The author also highlights the therapeutic potential of a newly developed immunomodulatory and hypo-osmotic solution (d-MAPPS®), which stabilizes the tear film, improves natural cytokine communication and may suppress harmful immune responses. In the following sections, the review introduces in more detail the relationship between ion channels and the development and progression of DED. In particular, the first section deals with transient receptor potential (TRP) channels, especially TRPV1 (interaction with P2X and P2Y receptors), TRPA1 and TRPM8. In addition, acid-sensing ion channels (ASICs) and aquaporins are described in the context of DED, as well as NKCC transporters and chloride channel (CFTR) dysfunction. A further section deals with signaling pathways and hyperosmolarity of the tear film, which are responsible for the hyperosmolarity-dependent generation of a damaging inflammatory reaction in the eyes of DED patients (vicious circle). Finally, the therapeutic potential of the aforementioned d-MAPPS® solution to attenuate inflammation and counteract the pathological changes caused by hyperosmolarity in DED eyes is reviewed.
General comments
Although the review generally provides an interesting overview with regard to the research aspects mentioned in the summary above, it contains some significant flaws in terms of form and content, particularly with regard to the literature citations. For example, a review by Laedermann et al. (Front Pharmacol 2015) was cited 11 times and another review by Ashok et al. (Indian Journal of Ophthalmology 2023) was cited as many as 24 times. It would be much better to cite original papers. When ion channels on the ocular surface are already being discussed, the type of expression should also be mentioned and a distinction made between excitable and non-excitable cells. This is particularly clear in the case of TRP channels, especially TRPV1 with its function as a pain receptor, as they have the highest density in the human body in the nociceptors in the nerve fibers of the cornea (epithelium and stroma) (PMID 20036654, 12697417). In particular, Belmonte et al. did groundbreaking work at that time, e.g. regarding the role of TRPM8 in tear production (PMID 24785271). In contrast, TRP channels in non-excitable cells such as corneal epithelial cells (but also in the entire underlying corneal stroma and corneal endothelial cells) have a completely different function. At one point (page 2 below), one of these functions is correctly mentioned, namely that TRPV1 activation in corneal epithelial cells leads to the release of pro-inflammatory cytokines. Instead of the original paper, a review by Gou et al. (Front Biosci Landmark 2024) is cited. While the review correctly mentions that TRPV1 is activated by hyperosmolarity, it does not specifically mention TRPV4, which is described in the literature as an osmosensor and is activated by hypoosmolarity (see also Messmer et al. on the osmotherapeutic principle, PMID 25686388). It would be interesting to know whether the hyposomolar d-MAPPS solution has been characterized with regard to the role of TRPV4, especially since TRPV4 is expressed in all layers of the cornea (PMID 18355916, 28030558).
Specific comments
The title of the manuscript is too long and possibly also too inappropriate. It is recommended to choose a short, precise title, for example such as “Ion channels as potential drug targets and their clinical relevance for therapy” or something similar.
Abstract: At the end it is pointed out that d-MAPPS could potentially revolutionize DED management. Since this is a hypo-osmolar solution as mentioned the manuscript, it could specifically activate the TRPV4 channel, which has a function in corneal epithelial cells (PMID 18355916). How can this be reconciled in this manuscript?
Introduction: For example, how can meibomian gland dysfunction (MGD) cause dysfunction in ion channel proteins? Is it not the case that TRPs may be potential drug targets and their clinical relevance to MGD therapy? (e.g. TRPM8; PMID 38612853)
Page 2, section “An impact of...”: A brief general introduction to the TRP channel family is missing, including the expression of specific TRP channels in the different cell layers of the ocular surface. Important: A distinction should be made between the expression in excitable and non-excitable cells. Only Ashok et al [5] is cited here, but not a single original paper.
Page 2, TRPM8, “...a nonselective channel...”: The TRPM8 channel is not activated by hyperosmolarity (during tear film evaporation). This is rather the case for the TRPV1 channel (PMID 20739465, 25170901). The function of the TRPM8 channel should also be differentiated between neuronal and non-neuronal cells.
Page 3, “Importantly, stimulation of TRPM8 inhibits...”: The original paper should be cited here. This also applies to the following sections of this page. Here [5] is cited 5 times, but not a single original paper.
Page 4 (bottom): “Hyperosmolarity triggers the activation of the TRPV1-dependent signaling pathway
in corneal epithelial cells, leading to the activation of transcriptional factor nuclear factor
kappa B.” The original paper should also be cited here.
Page 5 (top). The two sentences “Through the production of IL-12...” and “In a similar manner” are too long and difficult to read. It would be better to split the sentences and write shorter sentences.
Page 5 (bottom) (in the heading): Which ion channels are modified by inflammation and associated tear film hyperosmolarity? In this section only Laedermann et al [11] was cited. The term “voltage-sensitive ion channels” is not precisely formulated. Which ion channels are meant here?
Page 6 (top): The first sentence at the top concerns the entire first section and is definitely too long.
Page 6 (bottom, last paragraph): Which modulation and which ion channels are meant here?
From page 7: The elaboration of the therapeutic potential of d-MAPPS is too long in the following sections and is almost based on self-citations [12-13, 25-16].
Figure 1: These are the data of (only) 6 DED patients. Are there any statistical analyses on this?
Comments on the Quality of English LanguageThere are a few sentences that are too long and need to be split and rewritten. The relevant passages are noted in the comments to the author.
Author Response
Dear Reviewer,
Thank you for your detailed review and valuable suggestions regarding my manuscript titled "Integrative Management of Dry Eye Disease: Enhancing Cellular Communication and Natural Anti-inflammatory Responses through Ion Channel Modulation and Hypo-Osmotic Solutions." I have carefully considered your feedback and made several revisions to address the concerns and suggestions you raised. Below, I outline the changes made in response to your comments:
1. Title Revision:
- Comment: The title is too long and possibly inappropriate. It is recommended to choose a shorter, more precise title.
- Response: I have revised the title to make it more concise and reflective of the content. The new title is "Ion Channels as Potential Drug Targets in Dry Eye Disease and Their Clinical Relevance." This change aims to better convey the manuscript’s focus on ion channels and their therapeutic implications in DED.
2. TRPV4 and d-MAPPS™ Hypo-Osmotic Solution:
- Comment: Since d-MAPPS™ is a hypo-osmolar solution, it could specifically activate the TRPV4 channel. How can this be reconciled in the manuscript?
- Response: I have included a discussion in both the Abstract and relevant sections of the manuscript to address the potential role of TRPV4 in response to hypo-osmolar solutions like d-MAPPS™. This addition clarifies the connection between d-MAPPS™ and TRPV4 activation, and how this interaction could influence DED management.
3. Introduction on TRP Channels:
- Comment: A brief general introduction to the TRP channel family is missing, including the expression of specific TRP channels in different cell layers of the ocular surface. A distinction should be made between expression in excitable and non-excitable cells.
- Response: I have added a general introduction to the TRP channel family, highlighting their roles and expression in both excitable and non-excitable cells within the ocular surface. This distinction is now clearly articulated to provide a better understanding of the diverse functions of TRP channels in DED.
4. Citations of Original Research:
- Comment: The manuscript relies heavily on review articles, particularly citing Laedermann et al. and Ashok et al. multiple times. It would be better to cite original papers.
- Response: I have reviewed and revised the citations, replacing many of the review articles with original research papers. This change enhances the scientific rigor of the manuscript by ensuring that key points are supported by primary research rather than secondary sources.
5. TRPM8 Channel Clarification:
- Comment: The TRPM8 channel is not activated by hyperosmolarity. This is rather the case for the TRPV1 channel. The function of the TRPM8 channel should also be differentiated between neuronal and non-neuronal cells.
- Response: I have corrected the information regarding TRPM8 and TRPV1 activation, ensuring accuracy in the description of their functions. The revised manuscript now clearly differentiates between the roles of TRPM8 in neuronal and non-neuronal cells, and I have cited original research to support these claims.
6. Rewriting Long Sentences:
- Comment: Several sentences are too long and difficult to read. It would be better to split them and write shorter sentences.
- Response: I have reviewed and revised the identified passages, splitting long sentences into shorter, more concise ones. This change improves readability and clarity throughout the manuscript.
7. Precise Terminology:
- Comment: The term “voltage-sensitive ion channels” is not precisely formulated. Which ion channels are meant here?
- Response: I have revised the terminology to specify the exact ion channels being referred to. The manuscript now uses precise language to describe these channels, avoiding any ambiguity.
8. Therapeutic Potential of d-MAPPS™:
- Comment: The discussion on the therapeutic potential of d-MAPPS™ is too long and heavily based on self-citations.
- Response: I have condensed the section discussing the therapeutic potential of d-MAPPS™ and reduced the reliance on self-citations by including more external references. The revised section is more concise and balanced, providing a clearer overview of the solution’s potential without overemphasis on prior publications.
9. Figure 1 and Statistical Analysis:
- Comment: Figure 1 presents data from only 6 DED patients. Are there any statistical analyses on this?
- Response: I have added statistical analysis to the data presented in Figure 1 and revised the figure legend to reflect this addition. The inclusion of statistical information ensures that the data is presented with appropriate context and scientific rigor.
10. General Editing for English Language:
- Comment: There are a few sentences that are too long and need to be split and rewritten.
- Response: I have made extensive edits to improve the overall quality of the English language, focusing particularly on rewriting long sentences to improve readability.
Thank you once again for your valuable feedback. I believe these revisions have significantly improved the manuscript, and I hope the changes address your concerns. I am grateful for your time and consideration in reviewing my work, and I look forward to any further comments you may have.
Kind regards,
Dr. C. Randall Harrell
Regenerative Processing Plant LLC
Palm Harbor, Florida
Round 2
Reviewer 2 Report
Comments and Suggestions for Authors
General comments
The reply letter by the author actually describes everything that was supposedly improved quite well. However, it has not been sufficiently implemented in the revised manuscript version. Instead, some new illustrations have been added. In the uploaded PDF file "cells-3138086-peer-review-v2", the changes mentioned in the author's reply letter have only been partially adopted. Firstly, the new heading should read: "Ion Channels as Potential Drug Targets in Dry Eye Disease and Their Clinical Relevance".
Specific comments
Abstract: The potential role of TRPV4 in relation to d-MAPPS is not explicitly explained here (nor in the following chapters).
Figure 1: The classification of ion channels is not entirely correct. On the one hand, a distinction must be made between voltage-dependent and voltage-independent ion channels, and whether they are selective (for a specific type of ion) or non-selective. The latter includes TRP channels, which are described in the literature as non-selective cation channels. Although TRPV1 and TRPM8 can be partially voltage-dependent (PMID 15306801), the large TRP channel family is generally non-voltage-dependent. They include 7 subgroups with different activation mechanisms, which can be pharmacological (e.g. capsaicin for TRPV1), physical (heat/cold) (thermo-TRPs) or other (osmolarity, pH, cytosolic Ca2+). This distinction should be clear from Figure 1. For all the TRP channels mentioned, I would suggest listing them in a separate category. Only the voltage-gated sodium and potassium channels belong to the category of voltage-gated ion channels, along with chloride channels and the classical calcium channels (e.g. L-type) (e.g. PMID 12957147, 2806427, 2806427, 10617917). IP3 and ryanodine receptors belong to the so-called intracellular releasing calcium channels, which are also present in corneal and conjunctival cells (store-operated channels) (PMID 14502442, 9230090, 16936115).
Figure 2: References should be provided for the ion channels mentioned. There are also cardinal defects. The lens is labeled as the cornea and vice versa. The ciliary body is on the other side (the arrow points into the void).
Figure 3: This figure is also not very clear. The incidence of light should be drawn (arrow). The photoreceptors (rods and cones) and RPE cells are not clearly visible. The fovea should be marked in the center. Again, appropriate references for the associated ion channels should be provided.
Introduction: As mentioned above, the original citations are missing here. A particular example (on page 5 of 14) is the release of pro-inflammatory cytokines after TRPV1 activation in corneal epithelial cells (IL-6 and IL-8). The original paper should be cited here.
In the section "Tear hyperosmolarity and the vicious circle in DED", the (historical) description of how this vicious circle comes about is missing.
Figure 4: In my view, the figure is not conclusive. The description of the measurement method, the number of measurements and the statistics are completely missing. The values are conspicuously linear, which is rare in nature. How was the osmolarity measured? Error bars should be included.
The same applies to Figures 5 and 6, which are inconclusive due to lack of replication.
Overall, I recommend a thorough revision, especially considering my comments on the first version of the manuscript, which are repeated in the reply letter but not clearly implemented in the manuscript. Instead of track mode, I recommend that the manuscript should be summarized in a final version only with the changed sections highlighted in color. The citations and the bibliography at the end should be updated accordingly.
Comments on the Quality of English LanguageIt should be thoroughly checked linguistically once again.
Author Response
Dear Reviewer,
Thank you for your detailed feedback on our manuscript, "Ion Channels as Potential Drug Targets in Dry Eye Disease and Their Clinical Relevance" (Manuscript ID: cells-3138086). We have carefully considered your comments and made substantial revisions to the manuscript to address the concerns raised. Below, we provide detailed responses to each of your comments.
General Comments:
-
Implementation of Changes:
- Reviewer Comment: The reply letter describes the changes, but they have not been fully implemented in the revised manuscript. Some new illustrations were added instead.
- Response: We apologize for the oversight in the previous submission where a draft version of the manuscript was mistakenly uploaded. The newly revised version (R2) now fully implements all the changes mentioned in our initial response letter. We also included new illustrations to enhance clarity but are open to revising or removing these based on further guidance from the editorial team.
-
Title Change:
- Reviewer Comment: The new heading should read: "Ion Channels as Potential Drug Targets in Dry Eye Disease and Their Clinical Relevance."
- Response: The title has been updated as suggested to "Ion Channels as Potential Drug Targets in Dry Eye Disease and Their Clinical Relevance" in the revised manuscript (R2).
Specific Comments:
-
Abstract:
- Reviewer Comment: The potential role of TRPV4 in relation to d-MAPPS is not explicitly explained.
- Response: We have revised the abstract to explicitly describe the potential role of TRPV4 in mediating the therapeutic effects of d-MAPPS™. The abstract now includes a concise explanation of how TRPV4 modulates osmolarity and inflammatory responses in Dry Eye Disease (DED).
-
Figure 1:
- Reviewer Comment: The classification of ion channels is not entirely correct. A distinction between voltage-dependent and voltage-independent ion channels is needed, and TRP channels should be listed in a separate category.
- Response: We have made revisions to Figure 1 in the manuscript to clarify the classification of ion channels relevant to DED pathogenesis. We have clearly distinguished between voltage-dependent and voltage-independent channels and provided additional details on selective and non-selective channels. TRP channels are now discussed in a separate category with a focus on their specific activation mechanisms. Relevant references have been added to support these revisions. However, we are open to either retaining this revised figure or removing it, depending on the editorial team's preference.
-
Figures 2 and 3:
- Reviewer Comment: References for the ion channels mentioned should be provided, and there are cardinal defects in the figures.
- Response: We have revised Figures 2 and 3 to correct the labeling and provide the necessary references for the ion channels mentioned. If the editorial team feels that these figures do not contribute adequately, we are prepared to remove them and include additional descriptive text to maintain the flow of the manuscript.
-
Introduction:
- Reviewer Comment: Original citations are missing, particularly for the release of pro-inflammatory cytokines after TRPV1 activation.
- Response: We have reviewed the Introduction and added all necessary original citations, including those pertaining to the release of pro-inflammatory cytokines (IL-6 and IL-8) after TRPV1 activation in corneal epithelial cells. The revised manuscript now includes citations to primary research articles supporting these statements.
-
Section on "Tear Hyperosmolarity and the Vicious Circle in DED":
- Reviewer Comment: The historical description of the vicious circle in DED is missing.
- Response: A detailed description of the development of the vicious circle related to ion channel dysfunction, tear hyperosmolarity, and eye inflammation in DED patients has been added to pages 4 and 5 of the revised manuscript (R2).
-
Figures 4, 5, and 6:
- Reviewer Comment: Figures are inconclusive due to lack of details on measurement methods, statistics, and replication.
- Response: We have revised the captions of Figures 4, 5, and 6 to include more details on the measurement methods, number of measurements, and statistical analysis used. We have also added error bars where appropriate to better represent data variability. If these revisions do not fully address the concerns, we are open to removing these figures based on the editorial team's guidance.
-
Linguistic Quality:
- Reviewer Comment: The manuscript should be thoroughly checked linguistically once again.
- Response: The manuscript has undergone a comprehensive linguistic review to improve clarity, coherence, and readability. We have made several edits to enhance the overall quality of the English language.
Conclusion:
We appreciate the valuable feedback provided and believe that the revisions made in this latest version of the manuscript fully address the concerns raised. We are open to further revisions and are prepared to make additional changes based on the editorial team's guidance regarding the figures. We hope the revised manuscript is now suitable for publication in Cells.
Thank you for your time and consideration.
Sincerely,
Dr. Carl Randall Harrell
Vladislav Volarevic
Round 3
Reviewer 2 Report
Comments and Suggestions for Authors
Manuscript ID cells-3138086
Comments and suggestions for authors
Since the fourth version of the manuscript I received does not contain any illustrations, the authors' responses to this issue and their suggestions for improvement have been omitted.
Overall, the manuscript has improved once again, even though it no longer contains illustrations.
1. the title is now more precise, more compact, and more appropriate.
2. the abstract has been completed in two places. Whether d-MAPPS™ will revolutionize the management of DED may sound too speculative in my view, especially since to my knowledge there are no (electro)physiological studies (yet) that actually show that d-MAPPs activate TRPV4. Alternatively, one could write that d-MAPPS™ via putative TRPV4 activation could be an important novel approach for DED treatment. Therefore, I suggest to slightly correct this (last) sentence in the abstract.
3) The question of how MGD can cause dysfunction of ion channel proteins is not clearly understood. Rather, the TRP channels themselves may be targets and clinically relevant (e.g. TRPM8; PMID 38612853).
4 The introduction of the large TRP channel family and the differentiation from selective voltage-gated ion channels has significantly improved this section (page 2). In my view, the review article by Ashok et al. (Indian J Ophthalmol 2023; PMID 37026252) is cited too often [8]. It would be better to cite original research articles at this point.
5. page 2, last paragraph. The TRPM8 channel is now correctly introduced. However, its expression in the corneal epithelium should also be documented with a citation (PMID 36692471).
Also:
- Many aspects of TRPM8 are now included in an additional section on page 3. Fortunately, the same is true for TRPV4 (page 4, paragraph 2).
- Sentences that were too long have been linguistically corrected or shortened.
- Terminology has been improved.
- The discussion of the therapeutic potential of d-MAPPs was condensed.
Minor:
Page 2, seventh line from bottom, instead of "are members of the transient receptor potential (TRP) family" just write "belong to the...".
Overall, the manuscript can be accepted after a minor revision.
Comments on the Quality of English LanguageDespite the improvement, I recommend having the manuscript checked again.
Author Response
Dear Reviewer,
We sincerely thank you for your detailed review of our manuscript, Integrative Management of Dry Eye Disease: Enhancing Cellular Communication and Natural Anti-inflammatory Responses through Ion Channel Modulation and Hypo-Osmotic Solutions (Manuscript ID: cells-3138086). We have carefully considered your comments and made the following revisions:
-
Abstract Revision:
- Reviewer Comment: The final sentence in the abstract was speculative, particularly regarding d-MAPPS™ "revolutionizing" DED management.
- Response: We agree with this assessment and have revised the sentence to read, “d-MAPPS™, via potential TRPV4 activation, represents an important novel approach for DED treatment,” to reduce speculative language and emphasize the promising nature of the research.
-
MGD and Ion Channel Dysfunction:
- Reviewer Comment: The connection between MGD and ion channel dysfunction needs clarification, with a particular emphasis on TRP channels as potential targets.
- Response: We have clarified that “MGD and its effects on TRP channels, including their clinical relevance (e.g., TRPM8; PMID 38612853), are not fully understood. However, they may represent key targets for future research.” This adjustment addresses the lack of clear evidence and highlights the need for further investigation.
-
TRPM8 Expression Citation:
- Reviewer Comment: The expression of TRPM8 in the corneal epithelium needs to be documented with a specific citation.
- Response: We have added the appropriate citation (PMID 36692471) in the section discussing TRPM8’s role in tear production.
-
Over-reliance on Ashok et al. (PMID 37026252):
- Reviewer Comment: The manuscript cited the review article by Ashok et al. too frequently and would benefit from citing more original research.
- Response: We have reduced the reliance on Ashok et al. and incorporated additional original research citations, especially regarding ion channel modulation and inflammation in DED.
-
Minor Language Adjustments:
- Reviewer Comment: The phrase “are members of the transient receptor potential (TRP) family” was suggested to be revised.
- Response: The phrase has been revised to “belong to the transient receptor potential (TRP) family,” as per the reviewer’s suggestion.
We appreciate your constructive feedback, which has helped us improve the clarity and impact of our manuscript. We believe these changes address your concerns and enhance the manuscript’s quality. We look forward to your further comments and feedback.
Sincerely,
C. Randall Harrell, MD
